# Investigation of *Rickettsia conorii* in Patients Suspected of Having Crimean-Congo Hemorrhagic Fever

**DOI:** 10.3390/pathogens11090973

**Published:** 2022-08-26

**Authors:** Neda Baseri, Mostafa Salehi-Vaziri, Ehsan Mostafavi, Fahimeh Bagheri Amiri, Mina Latifian, John Stenos, Saber Esmaeili

**Affiliations:** 1Department of Epidemiology and Biostatics, Research Centre for Emerging and Reemerging Infectious Diseases, Pasteur Institute of Iran, Tehran 1316943551, Iran; 2National Reference Laboratory for Plague, Tularemia and Q Fever, Research Centre for Emerging and Reemerging Infectious Diseases, Pasteur Institute of Iran, Akanlu, KabudarAhang, Hamadan 6556153145, Iran; 3Department of Arboviruses & Viral Hemorrhagic Fevers (National Reference Laboratory), Pasteur Institute of Iran, Tehran 1316943551, Iran; 4Australian Rickettsial Reference Laboratory, University Hospital Geelong, Geelong, VIC 3220, Australia

**Keywords:** Mediterranean spotted fever, *Rickettsia conorii*, Congo-Crimean hemorrhagic fever, *Rickettsia helvetica*

## Abstract

*Rickettsia conorii* is the causative agent of Mediterranean spotted fever (MSF). Misdiagnosis of MSF may occur with febrile syndromes associated with rash and thrombocytopenia, such as Crimean-Congo hemorrhagic fever (CCHF). This study aimed to determine the prevalence of *R. conorii* among serum samples obtained from 260 suspected CCHF patients with features of MSF in Iran (2018–2020). The quantitative polymerase chain reaction (qPCR) method detected three (1.15%) positive 16S rDNA *Rickettsia* spp. samples that were classified as *R. conorii* subsp. *conorii*, *R. conorii* subsp. *Israelensis,* and *R. helvetica* using the sequencing of *gltA*, *ompA*, and *17kDa* genes. Furthermore, *R. conorii* IgM antibodies presented in 38 (14.62%) patients by the enzyme-linked immunosorbent assay (ELISA) method. Out of 97 MSF patients with available paired serum samples, IgM seroconversion and a four-fold increase were observed in 14 (14.43%) and 12 (12.37%) patients, respectively. We concluded that rickettsial agents are present in Iran and may be misdiagnosed with other febrile syndromes.

## 1. Introduction

*Rickettsia conorii*, an obligate intracellular coccobacillus, is known as the causative agent of Mediterranean spotted fever (MSF) disease [1]. MSF is a tick-borne disease and a member of the spotted fever group (SFG) rickettsioses [2]. In addition to *R. conorii*, MSF may occur by some other *Rickettsia* spp., including *Rickettsia. helvetica* [3]. MSF is usually accompanied by nonspecific symptoms, including fever, maculopapular erythematous rash, and a black eschar (tache noire) at the tick bite site [2]. Furthermore, leukopenia, thrombocytopenia, increased transaminases, and decreased serum ion (Na^+^, K^+^, and Ca^2+^) levels are common nonspecific laboratory findings in MSF [4]. Therefore, misdiagnosis may occur with other febrile syndromes associated with rash and thrombocytopenia. MSF mortality rates vary and depend on the geographic region and the duration of infection [5]. The high incidence and mortality of MSF are associated with warmer temperatures and low rainfall, which are favorable conditions for tick vector activity [6]. So far, four *R. conorii* subspecies, including *conorii*, *israelensis*, *caspia*, and *indicia,* have been identified. These subspecies are genetically similar but are different based on epidemiological and clinical characteristics [7]. In the life cycle of *R. conorii*, dogs and rodents are considered the reservoirs, while ticks (mainly *Ripcephalus sanguineus*) act as the vectors. Climatic conditions, specific vectors, and natural host restriction are the contributory factors for the endemicity of *R. conorii* in the Mediterranean region, including North Africa, southern Europe, and the Middle East. Limited data are available regarding rickettsial infections in Iran [8,9,10,11,12,13,14,15,16,17,18]. In a study conducted in 1995, a high proportion of positive human sera were identified by a rickettsial enzyme-linked immunosorbent assay (ELISA) (45%) kit in Iran. They showed that 27.5% of samples had antibodies against *R. conorii* using the immunofluorescent assay (IFA) test [8]. Recently, five MSF cases have been reported from Kerman province (southeast of Iran), with patients displaying symptoms of fever, rash, and anorexia. In four patients, black eschars and tick-bite history were observed [9]. It would appear that rickettsial disease is more prevalent in Iran, which is indicative of a knowledge gap in these diseases.

Most MSF patients (especially in fatal cases) have epidemiological features, clinical symptoms, and laboratory findings similar to Crimean-Congo hemorrhagic fever (CCHF) cases. There were reports of MSF among patients who were initially considered or suspected to be CCHF [4,19,20]. Almost 55% of probable CCHF cases in Iran that were sent to the National Reference Laboratory of Arboviruses and Hemorrhagic Fever, Pasteur Institute of Iran (NRLAHF) were CCHF negative [21]. Since nonspecific symptoms of MSF may be confused with CCHF, and there is no comprehensive information on the prevalence of this disease in Iran, we aimed to investigate the prevalence of *R. conorii* in CCHF suspected patients using molecular and serological methods.

## 2. Results

### 2.1. Demographic Features, Clinical Symptoms, and Laboratory Findings of Included CCHF-Negative Patients

In this study, 260 CCHF-negative serum samples (75.80% male and 24.20% female) were selected for further analysis (Table 1). The mean standard deviation (SD) and the median age of people were 44.02 (19.67) and 43.00 years, respectively. Among these 260 patients, 51.15% were ≤43 years, and 48.85% were >43 years.

In addition, 53.08% of included CCHF-negative patients lived in rural areas. The histories of contact with animals, tick bites, and current traveling were observed in 63.85%, 28.08%, and 4.23% of included patients, respectively.

Among CCHF-negatives, fever (90.38%), headache (41.15%), and Thrombocytopenia (95.83%) were common. Furthermore, increased levels of serum glutamic oxaloacetic transaminase (SGOT), serum glutamic pyruvic transaminase (SGPT), prothrombin time (PT), and partial thromboplastin time (PTT) were the predominant laboratory findings (Table 1). A history of contact with animals and tick bites was recorded from 166 urban and 73 rural patients.

Samples were collected from 25 different provinces of Iran (Table 2 and Figure 1). Most of the specimens were from Khorasan-Razavi (49 cases), Kerman (49 cases), Sistan and Baluchestan (30 cases), and Mazandaran (27 cases) provinces.

### 2.2. Seropositivity of R. conorii

Out of 260 patients, 38 cases (14.62%) were positive for *R. conorii* IgM. Evidence of seroconversion or ≥4-fold increase in *R. conorii* IgM antibody titers were observed in 14 (14.43%) and 12 (12.37%) patients of 97 patients with available paired serum samples, respectively. Of these, 8 (8.25%) patients showed 2 to 4-fold increased *R. conorii* IgM antibody titers. In addition, an intangible increase was detected in 4 (4.12%) paired samples (Table 3).

### 2.3. Molecular Identification of R. conorii

Out of 260 samples, 3 cases (1.15%) were positive for *Rickettsia* spp. *16S* rDNA gene by quantitative polymerase chain reaction (qPCR). Following the sequence analysis of the rickettsial genes examined, the patient codes 7051 and 7584 were identified as *R. conorii* infections with strain similarities with *R. conorii* subsp. *conorii*, and *R. conorii* subsp. *israelensis*, respectively. Patient 7051 resided in Joveyn County in Khorasan-Razavi (northeast of Iran) and had their blood drawn 7 days after the onset of clinical symptoms. Patient 7584 resided in the Jiroft County of Kerman provinces (southeast of Iran) and had their blood drawn 9 after the onset of clinical symptoms. In addition, a likely *R. helvetica* infection was detected in patient 7675 from Qazvin province, in which blood was drawn 5 days after the onset of clinical symptoms. In this study, phylogenetic analysis of positive molecular samples based on sequences of *Rickettsia gltA*, *ompA*, and *17kDa* genes is shown in Figure 2, Figure 3, and Figure 4, respectively. The *gltA* and *17kDa* genes were amplified in all 3 positive samples (7051, 7584, and 7675). The *ompA* was detected in samples 7051 and 7584 and was absent in sample 7675.

Both 7051 and 7584 samples were also positive for the *R. conorii* IgM antibody, while paired sera collected within a 5-day interval of patient 7675 with the *R. helvetica*-like infection were negative by *R. conorii* ELISA, indicative of a lack of cross-reaction between these two species. Comparison of changes in *R. conorii* IgM antibody titers in paired serum samples obtained from patients 7051 and 7584 showed a seroconversion accompanying a 3-fold and 4-fold increase in antibody titer.

### 2.4. Risk Factors

Among 25 exanimated provinces, MSF cases were isolated from 12 provinces (Table 2 and Figure 1). The highest number of positive cases came from Kerman (34.69%), Khorasan-Razavi (12.24%), and Sistan and Baluchestan (10%) provinces.

As shown in Table 1, the prevalence of MSF was higher in rural residents than in residents in towns (18.7% vs. 10.71%), but this difference was a correlation (*p* = 0.067, odds ratio (OR): 1.942, 95% confidence interval (95% CI): 0.95–4.01).

Among positive MSF cases, 20 (27.4%) patients had a history of a tick bite. Based on the statical analysis, a significant (*p* < 0.001) relation between tick bite history and MSF infection (OR: 3.54, 95% CI: 1.75–7.19) was also observed, and it remains statistically significant in multivariable modeling, also. There was a weaker statistical correlation (*p* = 0.08, OR: 2.00, 95% CI: 0.90–4.23) between MSF infection and contact with animals. Comparatively, contact with animals was more in MSF patients (17.5%) than in negative MSF cases (9.6%). There was no significant relationship between other characteristics and positive cases (See Table 1).

The present study showed that the prevalence of MSF in southern (33.33%, *p* = 0.012, OR: 10.17, 95% CI: 1.67–61.92) and southeastern (25.32%, *p* = 0.003, OR: 6.89, 95% CI: 1.94–24.42) regions of Iran compared to the northern region was significantly higher (Table 4).

## 3. Discussion

*R. conorii* causing MSF is endemic in the Mediterranean region, including northern Africa and southern Europe [1]. MSF was first described in 1910 in Tunisia and was soon reported in other regions around the Mediterranean Sea [22]. Twenty years later, it was identified as a rickettsial disease transmitted by the brown dog tick [23]. The region-specified disease is due to the climatic conditions and vector and natural host limitations [1]. Nowadays, MSF is likely to continue to be found in the expanded regions [1]. Sporadic cases have been found in more locations in Europe and Africa. Furthermore, imported cases were reported by international travels in the United States and northern Europe [1,24]. There are limited studies on rickettsial prevalence in Iran. Because of the misdiagnosis of MSF with CCHF, the present study described epidemiological features of MSF among patients with suspected CCHF and reported 14.62% seropositivity for *R. conorii* among these patients. MSF and CCHF diseases may present similar clinical and epidemiological characteristics, including fever, rash, petechiae, and eschar, thrombocytopenia, spread by tick bite, and contact with infected animals. MSF is caused by *R. conorii*, while CCHF is a viral disease. Therefore, differential diagnosis is important in the early treatment of MSF disease. In 2008 in Greece, a case of MSF expired due to misdiagnosis as CCHF and improper treatment [5,25]. In the present study, petechiae were also more common in MSF patients (25.90%) than in negative *R. conorii* patients (13.3%). In an endemic area for CCHF in Albania, 2.9% rickettsiosis and 29.4% leptospirosis were recognized among patients with suspected CCHF [19]. Furthermore, a coincidence between MSF and CCHF was reported in 2012 in an endemic area of CCHF in Turkey [25]. They showed that the seroprevalence of *R. conorii* was higher in CCHF patients (52.32%) than in CCHF negatives (25.82%). The first study from Iran in 1996 reported a high seropositive prevalence of SFG *rickettsiae* by ELISA (45.00%) and IFA (27.50%) tests [8]. In the present study, Kerman, Khorasan-Razavi, Sistan, and Baluchestan provinces had the highest prevalence of MSF infections (over 10%) (Figure 1). In 2017–2018, MSF cases were confirmed using the IFA test from Kerman province in southeast Iran [9]. Furthermore, molecular tests in ticks and human fleas confirmed *Rickettsia* spp. in the East Azarbaijan, Lorestan, and Zanjan provinces [10,11,12,13]. However, *Rickettsia* spp. was not detected among adult mosquitoes from Fars province in southern Iran in 2017 and 2018 and in the blood of camels in central and southeastern Iran in 2019 [16,18]. In contrast, *Rickettsia* spp. (including *R. monacensis*, *R. helvetica*, *R. heilongjiangensis*, *R. raoultii*, and *R. slovaca*) was detected from Ahvaz province (in the southwest) among 2.2% of blood samples isolated from dogs from five different regions of Iran [14]. In 2021, seroreactivity against *Rickettsia* spp. was identified in 1% of rats in urban areas of Tehran, Iran [17]. Therefore, it is possible that *Rickettsia* spp. could be circulating in different parts of Iran. Among European countries, Italy (particularly in Sicily), Spain, Portugal, and southern France have relatively high rates of *R. conorii*. These regions usually are classified as having a temperate Mediterranean climate [6,26]. MSF is also a prevalent disease in Turkey [27]. Since this is a neighboring country of Iran, *R. conorii* could be more prevalent than is currently known. In a study in Turkey, 77% of patients had a 4-fold increase in *R. conorii* antibodies in two sera collected within a 2-week interval using the IFA test. They also showed that 73% of skin biopsy samples tested positive for *ompA* and *gltA* gene PCR [27]. The previous studies determined that the suitable samples for molecular tests were skin biopsy from eschar and sometimes blood epidemiological and clinical characteristics [7]. In the present study, 3 (1.15%) serum samples were reported as *Rickettsia* spp. positive cases using the qPCR molecular test. Using a PCR assay for *Rickettsia* spp identification, 2 cases (0.77%) were confirmed as the agent of MSF (*R. conorii* subsp. *conorii* and *R. conorii* subsp. *Israelensis-*like) and 1 case as *R. helvetica*-like. *R. helvetica* was first recognized in 1979 in Switzerland from Ixodes ticks as a new member of SFG *Rickettsia* and is associated with an eruption fever in Europe and Asia [3,28,29]. Ticks and rodents act as the vector and reservoir of *R. helvetica*, respectively. Flu-like symptoms are the manifestations of the acute disease. In the chronic form, perimyocarditis may occur [30]. As we have potentially detected multiple SFG species in Iran, the healthcare network should have a wide radar for all SFG members.

Some factors such as age, gender, job, temperature, and climate may be linked to a higher *Rickettsia* infection rate [1]. Here, the highest MSF prevalence rate was observed in the southern and southeastern regions of Iran. These regions are characterized by hot and dry summers, similar to Mediterranean areas [31]. In the present study, the MSF cases were also significantly associated with a tick bite history, similar to the usual identifying factors for MSF. Seroepidemiological studies in Europe and Mediterranean countries have shown that *R. conorii* prevalence in people living in rural regions (10%-60%) is higher than in towns (2–30%) [25]. The village residents in the present study were more (but not statistically significant) infected than urban inhabitants (18.84% vs. 10.66%). The higher contact with ticks or tick-carrying animals in rural areas than in towns could be the reason for these differences. This survey showed that 66.67% of positive cases in urban areas had contact with animals. Therefore, in addition to rural areas, contact with animals in cities was also a risk factor for this disease because close contact with domestic animals carrying ectoparasites may result in exposure to rickettsial infections [32].

The limitations of the present study should be mentioned, including, (I) sample size in some provinces was not adequate to investigate the prevalence of the infection in those regions. The lack of positive MSF cases in some provinces could be due to the small number of samples in those provinces. Therefore, it is recommended that the epidemiological studies on *Rickettsia* in Iran be conducted with a larger sample size and wide geographical distribution. (II) Complete clinical and laboratory information did not exist because of a lack of accurate information recorded in the CCHF surveillance and detection system. Accurate recording of information at the time of admission or follow-up during hospitalization can be helpful. (III) Paired serum samples did not exist for 163 patients. To provide better results in serological changes at different time intervals, a second serum was needed for all 260 patients. (IV) IgG assessment was not undertaken to indicate the previous infection. (V) The difference between the serological and molecular results is due to the type of sample (serum) that was used in molecular detection. In the molecular test, serum samples could lead to false-negative results because the bacteria in rickettsial disease are not persistent long-term. In addition, the bacteria could be trapped in the blood clot and could not be detected in the separated serum by centrifuge. It is recommended that a skin biopsy or blood sample (in case of bacteremia) be used for molecular MSF diagnosis.

## 4. Materials and Methods

### 4.1. Sample Collection

In the present study, samples were selected from 1565 suspected CCHF patients that were referred from different regions of Iran to the NRLAHF from March 2018 to September 2020. Laboratory diagnosis of CCHF was performed using molecular and serological assays, as previously described [33].

Demographic, clinical, and laboratory data of these patients were collected from the available recorded data of CCHF in the NRLAHF. Based on this data, patients who were finally reported as CCHF negative and had criteria meeting the MSF case definition were selected for further investigations. The following items were considered as inclusion criteria: (I) having clinical symptoms of fever, skin rash, or eschar at the site of the bite; (II) epidemiological evidence, including geographical distribution, history of tick bites, and contact with animals; (III) laboratory findings such as leukopenia, thrombocytopenia, and increased liver transaminase enzymes. A total of 260 patients were selected based on these criteria, for which there were 97 patients with paired serum samples (with at least 3 days and a maximum of 7 days intervals). For the remaining patients (*n* = 163), single serum was available.

The case definition of MSF in this study was: I) confirmed MSF case: a clinically compatible case (meets clinical evidence criteria) along with one of the laboratory-confirmed criteria (includes elevated antibody titer agents *R. conorii* (between paired serum specimens) and detection of *R. conorii* DNA); II) Probable MSF case: a clinically compatible case (meets clinical criteria) with presumptive laboratory evidence, including a single positive antibody titer against *R. conorii*.

### 4.2. ELISA Serological Test

To determine the presence of IgM antibodies against *R. conorii*, selected serum samples were tested by the ELISA method using a commercial kit (*Rickettsia conorii* ELISA IgM kit, Vircell, Granada, Spain), as per the manufacturer’s instructions. For paired serum samples, both the first and second sera were tested simultaneously.

### 4.3. DNA Extraction and TaqMan q PCR

Genomic DNA from all sera samples was extracted using the Mammalian Blood DNA Isolation Kit (Viragene., Tehran, Iran) according to the manufacturer’s instructions. The extracted DNA was stored at −20 °C for future molecular analysis.

All extracted DNA samples were tested by qPCR to detect the *16S* rDNA gene of *Rickettsia* spp. using the forward primer (5’-CGCAACCCTYATTCTTATTG-3’), reverse primer (5’-CCTCTGTAAACACCATTGTAGCA-3’), and probe (6-FAM-TAAGAAAACTGCCGG-TGATAAGCCGGAG-TAMRA) [34]. The reagents and equipment used were RealQ Plus 2x Master Mix (Ampliqon, Copenhagen, Denmark) and Corbett 6000 Rotor-Gene system (QIAGEN, Hilden, Germany). *R. conorii* genomic DNA (Vircell, Granada, Spain) and distilled water were used as the positive and negative controls, respectively. Quantitative analysis of the results was performed using Rotor-Gene Q^®^ Q 2.3.5 software (QIAGEN, Hilden, Germany).

### 4.4. Rickettsia Species Identification and Phylogenetic Analysis

*Rickettsia* species identification and phylogenetic analysis of *16S* rDNA positive samples were performed by amplification of the *gltA*, *ompA*, and *17kDa* genes using conventional PCR methods. Previously described primers [35,36] and 2x TEMPase Hot Start Master Mix BLUE A (Ampliqon, Copenhagen, Denmark) were used in PCR reaction mixtures and PCR products sequenced externally (Genomin, Tehran, Iran).

The *gltA*, *17kDa*, and *ompA* gene sequences of the various rickettsial species were sourced from GenBank and underwent phylogenetic analysis using the sequences generated from our study using MEGA software (version MEGA X 10.1, Mega Limited, Auckland, New Zealand).

### 4.5. Statistical Analysis

For statistical analyses of data, SPSS software (version 23, SPSS Inc., Chicago, IL, USA) was used. Descriptive analysis was used for qualitative and quantitative as frequency (and %) and mean ± SD. Univariable and multivariable logistic regressions were used to analyze risk factors associated with infection, and corresponding ORs and 95% CIs were determined. Variables from univariate analyses with *p* < 0.20 were recruited for the multivariable logistic regression model. Multivariable logistic regressions were performed in a backward stepwise approach with results from the best model selected by Hosmer–Lemeshow, and the final model is reported in this study. The two-sided statistical significance level, *p*-value, was set at 0.05 for all analyses in this study.

## 5. Conclusions

In conclusion, our data confirm that MSF could be a significant health problem in Iran that is far from the view of physicians and laboratories. Therefore, the lack of studies on MSF in Iran does not indicate the absence of this disease in the country. The study also shows that patients who are considered as suspected CCHF could likely be suspected of *R. conorii*. Since *R. helvetica* was also identified using the molecular test, it is recommended that physicians and laboratories should be aware of the presence of MSF and likely other rickettsial diseases in Iran. Therefore, studies for the causative organisms of this disease should be performed among a wide range of organisms to determine the host and reservoir ranges, pathogenesis, and the exact prevalence of MSF and other SFGs in Iran. Furthermore, MSF infection could be more common in southern Iran, especially in rural areas in patients with a tick bite history and contact with animals. The health system should be more aware of the possibility of MSF infection in these areas and assess the mentioned factors as epidemiological determinants of the disease in Iran.

## Figures and Tables

**Figure 1 pathogens-11-00973-f001:**
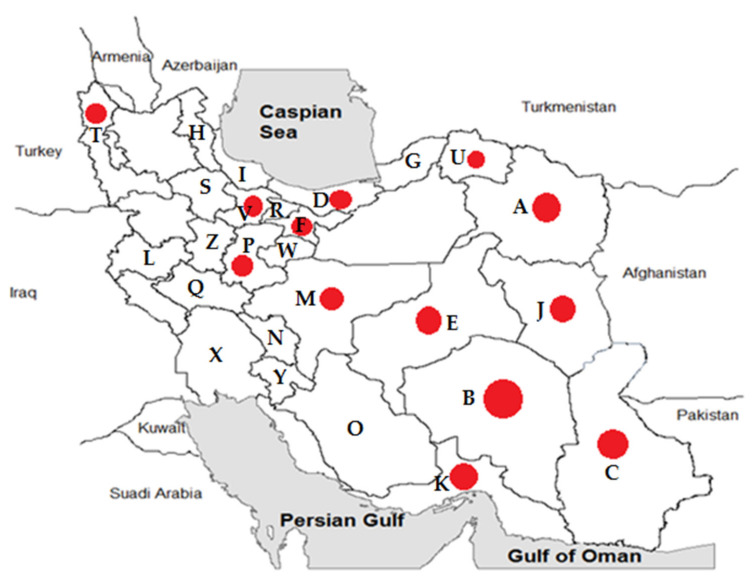
Geographic distribution of positive Mediterranean spotted fever cases (14.61%) in provinces of Iran, 2018–2020. The sampling of rodents was conducted in Khorasan-Razavi (A), Kerman (B), Sistan and Baluchestan (C), Mazandaran (D), Yazd (E), Tehran (F), Golestan (G), Ardabil (H), Gilan (I), South-Khorasan (J), Hormozgan (K), Kermanshah (L), Isfahan (M), Chaharmahal and Bakhtiari (N), Fars (O), Markazi (P), Lorestan (Q), Alborz (R), Zanjan (S), West-Azerbaijan (T), North-Khorasan (U), Qazvin (V), Qom (W), Khuzestan (X), Kohgiluyeh and Boyer-Ahmad (Y), and Hamedan (Z) provinces of Iran. Red circles indicate positive MSF cases in those provinces.

**Figure 2 pathogens-11-00973-f002:**
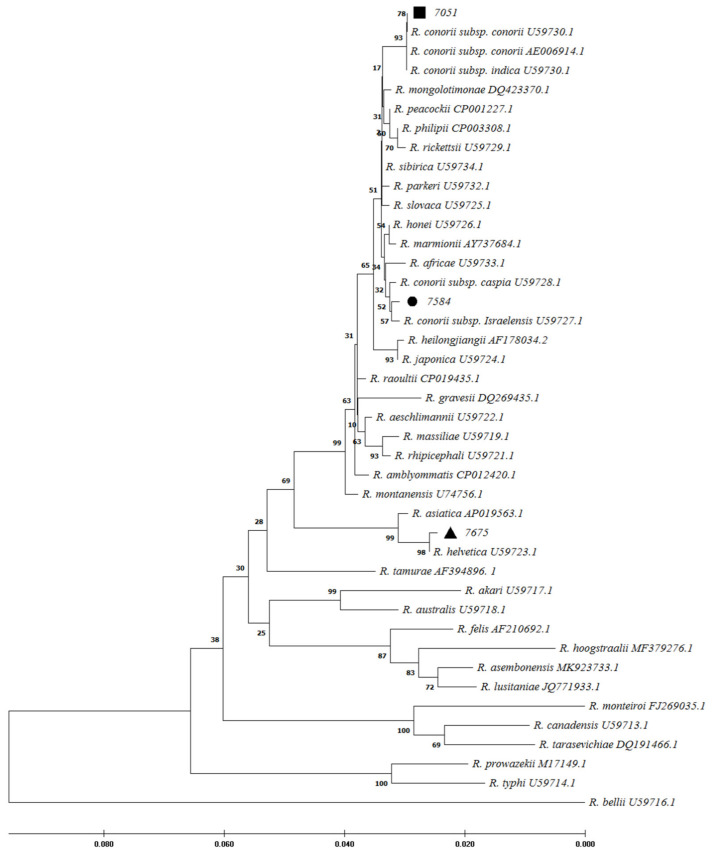
Phylogenetic 
analysis based on *Rickettsia*
*gltA* gene sequencing and Maximum 
Likelihood method algorithm (Kimura 2-parameter model). The tests were 
performed with bootstrap (1000 repetitions) by MEGA X10.1 software. Sample IDs 7584, 
7051, and 7675 were the samples studied in this study and aligned closely to *Rickettsia 
conorii *subsp. *israelensis*, *R. conorii* subsp. *conorii*, and *R. helvetica*, respectively.

**Figure 3 pathogens-11-00973-f003:**
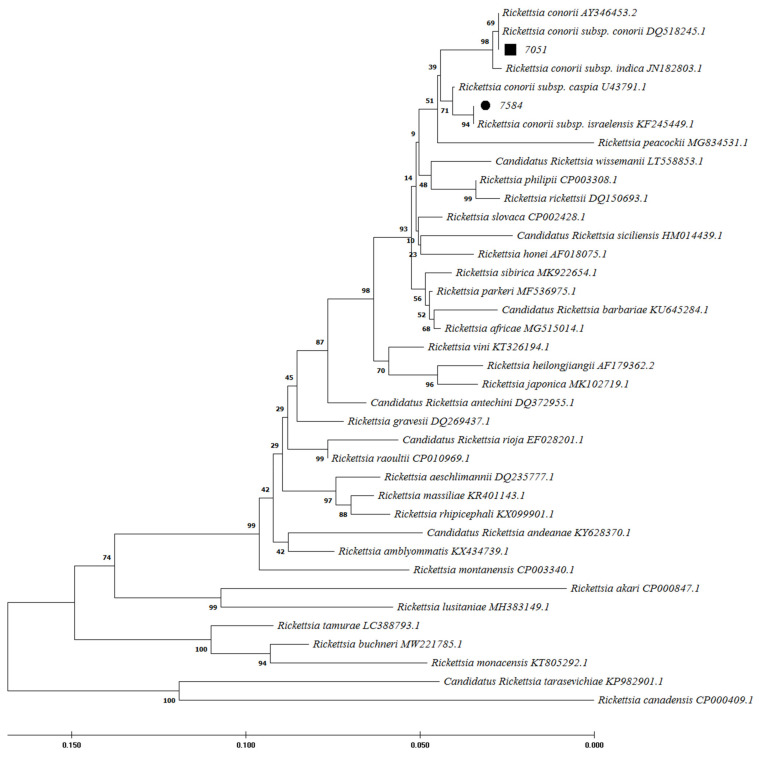
Phylogenetic analysis based on *Rickettsia ompA* gene sequencing and Maximum Likelihood method algorithm (Kimura 2-parameter model). The tests were performed with bootstrap (1000 repetitions) by MEGA X10.1 software. Samples 7584 and 7051 were the samples studied in this study with the closest homology to *Rickettsia conorii* subsp. *israelensis*, and *R. conorii* subsp. *conorii*, respectively. The *ompA* gene was not amplifiable in sample 7675.

**Figure 4 pathogens-11-00973-f004:**
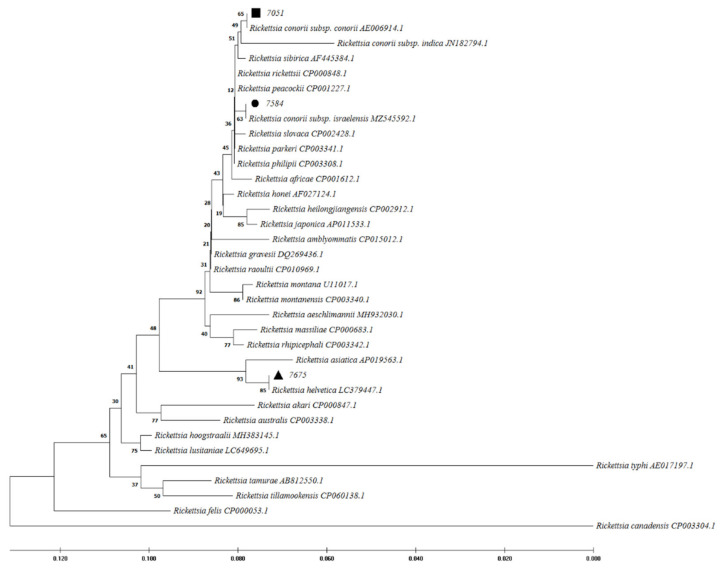
Phylogenetic analysis based on *Rickettsia 17kDa* gene sequencing and Maximum Likelihood method algorithm (Kimura 2-parameter model). The tests were performed with bootstrap (1000 repetitions) by MEGA X10.1 software. Sample 7584, 7051, and 7675 aligned closely to *Rickettsia conorii* subsp. *israelensis*, *R. conorii* subsp. *conorii*, and *R. helvetica*, respectively.

**Table 1 pathogens-11-00973-t001:** Univariable and multivariable logistic regression analysis of potential risk factors associated with *Rickettsia conorii* infection in Crimean-Congo hemorrhagic fever -negative patients based on demographic features, clinical symptoms, and laboratory findings.

Variables	Subgroup	*Rickettsia* Test Result		Univariable Analysis		Multivariable Analysis	
		No. (%) of neg	No. (%) of pos	*p*-Value	OR * (95% CI ^¥^)	*p*-Value	Adjusted OR * (95% CI ^¥^)
**Age**	43 ≥ years	114 (85.72%)	19 (14.28%)	0.74	0.89 (0.45–1.8)	−	−
	43 < years	107 (84.25%)	20 (15.75%)				
**Gender**	Female	54 (85.7)	9 (14.3)	0.93	1.04 (0.46–2.32)	−	−
	Male	168 (85.3)	29 (14.7)				
**Living place**	Urban	109 (89.34)	13 (10.66)	0.067	1.94 (0.95–4.01)	−	−
	Rural	112 (81.16)	26 (18.84)				
**Contact with animals**	No	85 (90.4)	9 (9.6)	0.08	2.00 (0.90–4.23)	−	−
	Yes	137 (82.5)	29 (17.5)				
**Tick bite/** **Tick contact**	No	169 (90.4)	18 (9.6)	<0.001	3.54 (1.75–7.19)	0.001	3.46 (1.70–7.05)
	Yes	53 (72.6)	20 (27.4)				
**Traveling**	No	213 (85.5)	36 (14.5)	0.67	1.32 (0.27–6.34)	−	−
	Yes	9 (81.8)	2 (18.2)				
**Clinical signs/symptoms**							
**Fever**	No	22 (88.0)	3 (12.0)	0.99	1.33 (0.38–4.67)	−	−
	Yes	200 (85.1)	35 (14.9)				
**Anorexia**	No	184 (86.8)	28 (13.2)	0.18	1.73 (0.76–3.86)	−	−
	Yes	38 (79.2)	10 (20.8)				
**Headache**	No	134 (87.6)	19 (12.4)	0.23	1.52 (0.76–3.04)	−	−
	Yes	88 (82.2)	19 (17.8)				
**Asthenia**	No	163 (86.2)	26 (13.8)	0.52	1·28 (0·61–2·67)	−	−
	Yes	59 (83.1)	12 (16.9)				
**Cough**	No	220 (85.6)	37 (14.4)	0.38	2.97 (0.26–33.26)	−	−
	Yes	2 (66.7)	1 (33.3)				
**Myalgia**	No	134 (85.4)	23 (14.6)	0.99	0.99 (0.49–2.01)	−	−
	Yes	88 (85.4)	15 (14.6)				
**Icterus**	No	218(85.2)	38 (14.8)	0.99	ND	−	−
	Yes	4 (100.0)	0 (0.0)				
**Abdominal pain**	No	191 (84.1)	36 (15.9)	0.19	0.34 (0.08–1.49)	0.19	0.37 (0.08–1.65)
	Yes	31 (93.9)	2 (6.1)				
**Nausea**	No	184 (85.6)	31 (14.4)	0.84	1.09 (0.45–2.70)	−	−
	Yes	38 (84.4)	7 (15.6)				
**Hemorrhage**	No	194 (86.2)	31 (13.8)	0.33	1.57 (0.63–3.89)	−	−
	Yes	28 (80.0)	7 (20.0)				
**Epistaxis**	No	216 (85.0)	38 (15.0)	0.60	ND	−	−
	Yes	6 (100.0)	0 (0.0)				
**Hematuria**	No	199(86.5)	31 (13.5)	0.17	1.95 (0.77–4.94)	−	−
	Yes	23 (76.7)	7 (23.3)				
**Petechiae**	No	202 (86.7)	31 (13.3)	0.08	2.28 (0.89–5.84)	−	−
	Yes	20 (74.1)	7 (25.9)				
**Melena**	No	216 (85.4)	37 (14.6)	0.99	0.97 (0.11–8.32)	−	−
	Yes	6 (85.7)	1 (14.3)				
**Diarrhea**	No	211(85.4)	36 (14.6)	0.99	1.07 (0.23–5.01)	−	−
	Yes	11 (84.6)	2 (15.4)				
**Chill**	No	187 (85.4)	32 (14.6)	0.99	1.01(0.39–2.57)	−	−
	Yes	35 (85.4)	6 (14.6)				
**Unconsciousness**	No	200 (84.7)	36 (15.3)	0.55	0.51 (0.11–2.24)	−	−
	Yes	22 (91.7)	2 (8.3)				
**Convulsion**	No	214 (85.3)	37 (14.7)	0.99	0.72 (0.09–5.95)	−	−
	Yes	8 (88.9)	1 (11.1)				
**Hemoglobin concentration**	Decreased	29 (93.5)	2 (6.5)	0.28	2.71 (0.62–11.84)	−	−
	Normal	193 (84.3)	36 (15.7)				
**Thrombocytopenia**	No	9 (90.0)	1 (10.0)	0.99	1.56 (0.19–12.72)	−	−
	Yes	196 (85.2)	34 (14.8)				
**Leucopenia**	No	76 (86.4)	12 (13.6)	0.96	0.97 (0.40–2.40)	−	−
	Yes	65 (86.7)	10 (13.3)				
**Leukocytosis**	No	107 (85.6)	18 (14.4)	0.54	0.70 (0.22–2.21)	−	−
	Yes	34 (89.5)	4 (10.5)				
**Proteinuria**	No	8 (88.9)	1 (11.1)	0.99	ND	−	−
	Yes	7(100.0)	0 (0.0)				
**PT**	Increase	21 (95.5)	1 (4.5)	0.99	1.612 (0.16–16.51)	−	−
	Normal	39 (92.9)	3 (7.1)				
**PTT**	Increase	25 (89.3)	3 (10.7)	0.99	0.78 (0.15–4.21)	−	−
	Normal	32 (91.4)	3 (8.6)				
**SGOT**	Increase	43 (81.1)	10 (18.9)	0.59	ND	−	−
	Normal	6 (100.0)	0 (0.0)				
**SGPT**	Increase	43 (81.1)	10 (18.9)	0.58	ND	−	−
	Normal	6 (100.0)	0 (0.0)				

*: Odds ratio, ^¥^: 95% confidence interval, ND: not defined, PT: prothrombin time, PTT: partial thromboplastin time, SGOT: serum glutamic oxaloacetic transaminase, SGPT: serum glutamic pyruvic transaminase.

**Table 2 pathogens-11-00973-t002:** Geographic distribution of positive Mediterranean spotted fever (MSF) cases in Iran, 2018–2020.

Province	No. of Patients	No. (%) of Positive MSF Cases
Khorasan-Razavi	49	6 (12.24)
Kerman	49	17 (34.69)
Sistan and Baluchestan	30	3 (10)
Mazandaran	27	2 (7.41)
Yazd	16	1 (6.25)
Tehran	14	1 (7.14)
Golestan	12	0 (0)
Ardabil	11	0 (0)
Gilan	8	0 (0)
South-Khorasan	6	1 (16.67)
Hormozgan	5	3 (60)
Kermanshah	5	0 (0)
Isfahan	4	1 (25)
Chaharmahal and Bakhtiari	4	0 (0)
Fars	4	0 (0)
Markazi	2	1 (50)
Lorestan	2	0 (0)
Alborz	2	0 (0)
Zanjan	2	0 (0)
West-Azerbaijan	2	1 (50)
North-Khorasan	1	1 (100)
Qazvin*	1*	0 (0)
Qom	1	0 (0)
Khuzestan	1	0 (0)
Kohgiluyeh and Boyer-Ahmad	1	0 (0)
Hamedan	1	0 (0)
Total	260	38 (14.61)

* Molecular positive for *Rickettsia helvetica*.

**Table 3 pathogens-11-00973-t003:** Comparison of changes in *Rickettsia conorii* IgM antibody titers with paired serum samples (*n* = 97).

Serological Criteria	No. of Patients with Increased *R. conorii* IgM Antibody Titers (%)
Seroconversion	14 (14.43)
≥4-fold increased *R. conorii* IgM antibody titers	12 (12.37)
3- to 4-fold increased *R. conorii* IgM antibody titers	2 (2.06)
2- to 3-fold increased *R. conorii* IgM antibody titers	6 (6.18)
Positive but intangible increase	4 (4.12)
Negative	73 (75.26)

**Table 4 pathogens-11-00973-t004:** Logistic regression analysis of the relationship between different geographic regions of Iran and Mediterranean spotted fever patients.

Regions (In Iran)	Included Provinces	No. of Samples (%)	No. of Positive Samples (Prevalence %)	*p*-Value	OR ^*^ (95% CI ^¥^)
North	Alborz, Qazvin, Guilan, Mazandaran, Golestan, Tehran	64 (24.61)	3 (4.69)	Reference	-
North-West	Ardabil, West-Azerbaijan, Zanjan	15 (5.77)	1 (6.67)	0.75	1.45 (0.14–15.02)
West	Hamadan, Kermanshah, Khuzestan, Kohgiluyeh and Boyer-Ahmad, Lorestan	10 (3.84)	0 (0.00)	0.99	ND
Central	Isfahan, Chaharmahal and Bakhtiari, Markazi, Qom, Yazd	27 (10.38)	3 (11.11)	0.27	2.54 (0.48–13.48)
South-Eastern	Kerman, Sistan and Baluchestan	79 (30.38)	20 (25.32)	0.003	6.89 (1.94–24.42)
South	Hormozgan, Fars	9 (3.46)	3 (33.33)	0.012	10.17 (1.67–61.92)
North-East	Khorasan-Razavi, South- Khorasan, North- Khorasan	56 (21.54)	8 (14.29)	0.08	3.39 (0.85–13.47)

*: Odds ratio, ^¥^: 95% confidence interval, ND: Not defined.

## Data Availability

Data that support the findings of this study are available within the article and from the corresponding author upon request.

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
