# Peer review of "Investigation of Rickettsia conorii in Patients Suspected of Having Crimean-Congo Hemorrhagic Fever"

_pathogens, 2022, doi:10.3390/pathogens11090973_

Round 1

Reviewer 1 Report

A nice descriptive study on the potential of misdiagnosing Crimean-Congo hemorrhagic fever (CCHF) with Mediterranean spotted fever (MSF). Serum samples were obtained from 260 suspected CCHF patients with features of MSF in Iran (2018-2020). Main finding was the confirmation of the existence of MSF in Iran and that patients exhibiting CCHF symptoms might have MSF. The paper is well written and organized and my comments are fairly minor.

Specific comments:

Introduction

L. 42. Italicize “indica”

L. 47-48. Add “and the Middle East”

L. 49 insert space “in1995”

L. 51 define “IFA”

L. 63-64. Authors say the aim was to “investigate the incidence of R. conorii” . I am assuming this would be back-calculated as a case-control design. But this is not stated anywhere explicitly and not in the data analysis. In that case, explain more thoroughly the criteria used for selecting the “controls”. Otherwise, this is not clear. Clearly “incidence” was not measured directly but appears to be inferred from prevalence and OR calculations. Any case, this aspect of the term “incidence” used here is confusing and needs to be much better explained and clarified. It appears using “incidence” here is misleading as you are really comparing prevalences with study design being “cross sectional”.

Results:

2.1. “Characteristic of demographic, clinical, and laboratory of samples” – some grammatic issues with this sentence structure.

L. 71-2. Title of Table 1 should indicate it refers to “risk factors” associated with infection risk. Important so it is easy to refer with the relevant part in the text. Table 1 title also does not provide the information needed to understand the data. For example, indicate statistical test used for p-values, define OR (and in the stats section in method explain how the OR was calculated).

L. 75. I am suggesting to move the “Risk factor” section immediately after Table 1 so that the discussion of the risk factors related directly to the significant ORs in table 1

L. 78. Space from166

Table 2. Indicate significant differences in prevalences between sites. Also, strongly suggest to add a map indicating sites /areas of higher prevalences.

L. 89. Table 3 is called out before there is a call-out for Table 2. Either change order in the text or change the order of the tables. Table 2 is only called out later in section 2.4

L. 101. Qzvin provice”  - without a map that doesn’t mean anything to most of us

L 101. R. Helvetica – it is never mentioned anywhere which disease it causes

L 110. Lowercase R. Canary

Section 2.4. Needs some more work. The statistical approach to refer ORs is not clear. There should be direct correspondence between the text here and the ORs in Table 1

Discussion

Discussion should remind the reader more thoroughly about the nature history of these pathogens so one can related the ecology of the system with the risk factors identified. Under which setting is it suspected that most exposure takes place.

L155. What is “petechiae”?

L. 169. Is there any suspected interactions between CCHF and MSF (increasing risk for one because of the other?)

L 163. Again – map would be very helpful for this to make sense

L. 173. Any speculations for what underlies these geographical patterns (environment. Climate, social factors?)

I like the discussion in L. 191-198 – giving ecologic context to exposure risk

Author Response

Dear Reviewer 1,

We appreciate your time and effort dedicated to providing feedback and detailed and thoughtful comments on our manuscript.

Please see the attachment,

Best Regards,

All authors

Reviewer 2 Report

Thank you for undertaking this exploration of the relationship between Mediterranean Spotted Fever seroprevalence among suspect CCHF patients. There is a lot of useful and interesting information in the piece, but to best address the purported goal, modification is appropriate.

-        The largest issue is the lack of incorporation of results from CCHF + results. Antibody rise appears to have been present in a minority of instances. If arguing for clinical discernment based upon this data, the possibility of prior and concomitant infection should be scaled.

-        Age should be an expressly described factor so age cohort effect can be considered.

-        Does table 1 reflect univariate or multivariate determinations of estimate of effect? Single or two tailed? And, the antibody rise subset should be observed separately.

-        Phylogenetic trees… there are only 3 sequences, should be visually highlighted and perhaps discussed in a single tree.

-        Under risk factors, the results should be couched by analytic approach (where interactions among factors are present, confounding, …).

-        Discussion is interesting, though should reflect on the changes discussed here, as ultimately should the conclusions. Currently the conclusions are unfocused on use case from the information.

-        Methods 4.2, No IgG assessment was undertaken?

-        Methods 4.5 should address the analytic aspects above flagged.

Author Response

Dear Reviewer 2,

We appreciate your time and effort the reviewer dedicated to providing feedback and detailed and thoughtful comments on our manuscript.

Please see the attachment,

Best Regards,

All authors

Round 2

Reviewer 2 Report

Thank you for carefully addressing the comments. The transparency added helps.